

# A NOVEL SINGLE-CAVITY MULTI-WAVELENGTH PHOTOACOUSTIC SPECTROMETER FOR ATMOSPHERIC AEROSOL RESEARCH

Claudia Linke[1], Inas Ibrahim[1], Nina Schleicher[2], Regina Hitzenberger[3], Meinrat O. Andreae[4], Thomas Leisner[1] and Martin Schnaiter[1]

[1]Institute of Meteorology and Climate Research, Atmospheric Aerosol Research, KIT, Germany
[2]Institute of Geography and Geoecology, KIT, Germany
[3]University of Vienna, Faculty of Physics, Austria
[4]Biogeochemistry Department, Max Planck Institute for Chemistry, Mainz, Germany

*Correspondence to: Claudia Linke (Claudia.Linke@kit.edu)*

**Abstract:** The spectral light absorbing behavior of carbonaceous aerosols varies depending on the chemical composition and structure of the particles. A new single-cavity multi-wavelength photoacoustic spectrometer was developed and characterized for measuring absorption coefficients at three wavelengths across the visible spectral range. In laboratory studies, several types of soot with different organic content were generated by a diffusion flame burner and were investigated for changes in mass-specific absorption cross sections (MAC), absorption and scattering Ångström exponents ($\alpha_{abs}$ and $\alpha_{sca}$), and single scattering albedo ($\omega$). By increasing the organic carbonaceous (OC) content of the aerosol from 50% to 90% of the total carbonaceous mass, for 660 nm nearly no change of MAC was found with increasing OC content. In contrast, for 532 nm a significant, and for 445 nm a strong increase of MAC was found with increasing OC content of the aerosol. Depending on the OC content, the Ångström exponents of absorption and scattering as well as the single scattering albedo increased. These laboratory results were compared to a field study at a traffic dominated urban site, which was also influenced by residential wood combustion. For this site a daily average value of $\alpha_{abs}$(445-660) of 1.9 was found.

## 1. INTRODUCTION

Carbonaceous particulate matter is a unique component of the atmospheric aerosol because of its absorption ability and large chemical variability (IPCC 2013). Its presence in the atmosphere causes direct and indirect climatic effects. The ability to absorb and scatter sunlight over a wide spectral range directly affects Earth's radiative balance. Indirectly, the presence of aerosol particles influences the life time of clouds and their precipitation properties (Rosenfeld et al. 2008, Sun et al. 2007).

In climate modeling, carbonaceous aerosols are represented as black carbon (BC) and organic carbon (OC; ICCP 2013). In contrast to BC, which is a strongly light absorbing material, most models assume OC to be a non-absorbing material (Kameel et al. 2014, Andreae & Ramanathan 2013, Chung et al. 2012). Based on these assumptions, BC contributes by a rather strong positive forcing and OC by a negative forcing to the overall radiative forcing by carbonaceous aerosols, which is then estimated to be close to zero (IPCC 2013).



While BC is of deep black appearance, there are further carbonaceous compounds that contribute to atmospheric aerosol absorption. Humic-like substances (HULIS) or brown carbon (brC) are of yellow or brownish color. Even though these compounds are only weak absorbers in comparison to BC, their steeply increasing light absorption towards the UV and
their atmospheric abundance might result in a significant contribution to the shortwave absorption by aerosols in the atmosphere (Kirchstetter et al. 2012, Chakrabarty et al. 2010, Hoffer et al. 2006, Andreae & Gelencsér 2006).

The formation of absorbing organic aerosol particles can occur through different pathways. Chemical and photochemical processes as well as coagulation and oligomerization
potentially end up in macromolecular structures. Such chemical systems will form multiple conjugated double and aromatic bonds as well as polyfunctional groups, which build light absorbing chromophoric structures. These chromophoric structures are responsible for the wavelength-dependent light absorption by organic species in the visible and near UV spectral range (Rincón et al., 2009; Prévôt et al., 2009; Gelencsér et al., 2002).

The transition from volatile organic carbon (VOC) to semi volatile (SVOC), low volatile (LVOC), and extremely low volatile (ELVOC) carbon is continuous. Saleh et al. (2014) showed that for biomass burning emissions an increase in light absorptivity is linked to a decrease in volatility of the organic compounds.

Emissions from biomass burning and to a large extent from fossil- and biofuel combustion
contain both black carbon and organic carbon. During burning processes, carbonaceous aerosols are generated by incomplete combustion of these fuels. The chemical composition of the resulting particles is strongly dependent on the combustion conditions (Andreae & Gelencsér 2006). Particularly, biomass and wood burning is often performed at low temperatures, resulting in high emissions of light absorbing particulate matter (Kirchstetter
2004; Sandradewi 2008). Moreover, co-emitted primary organic matter (POM) of biomass burning emissions might be weakly or even non-absorbing, but enhances the absorbing properties when forming an internal mixture with BC. Therefore, the spectral light absorption properties of such internally mixed carbonaceous particles may be shifted toward the UV wavelength range (Lack et al. 2012).

Considering the diversity of possible sources for carbonaceous particulate matter in the atmosphere, which covers the whole range from biogenic emissions over wildfires to residential heating, it is clear that the contribution of carbonaceous material to climate forcing on a global scale still remains difficult to estimate (Kirchstetter et al. 2012; Bond et al., 2013; Andreae & Ramanathan, 2013).

For BC, the imaginary part of the refractive index is nearly wavelength independent over the visible and near-UV spectral range. In contrast, the imaginary part of the refractive index of brown carbon (brC) increases towards shorter wavelengths (Moosmüller et al. 2009, Schnaiter et al. 2006). The small size of the particles and their spectral refractive index induce a wavelength dependence of the aerosol absorption coefficient, which is
characterized by the absorption Ångström exponent, $\alpha_{abs}$. While, in the case of BC, the absorption coefficient is only slightly increasing with decreasing wavelength, which results in $\alpha_{abs}$ close to 1.0 (Lewis et al. 2008, Kirchstetter et al. 2004, Schnaiter et al. 2003), brC can have a significantly higher absorption coefficient in the blue and near-UV spectral range





resulting in an $\alpha_{abs}$ higher than 1. From laboratory (McMeeking et al. 2014, Schnaiter et al. 2006) and field studies (Kirchstetter & Thatcher 2012, Rizzo et al. 2011, Sandradevi et al. 2008) there is evidence that for brC the $\alpha_{abs}$ is much larger than 1.0.

McMeeking et al. (2014) reported values for the absorption Ångström exponent $\alpha_{abs}$ between
1.5 and 7 from laboratory biomass burning experiments using a variety of different fuels. For carbon particles generated in a diffusion flame under different burning conditions, Schnaiter et al. (2006) determined $\alpha_{abs}$ values in the range of 1 to 7.

Kirchstetter & Thatcher (2012) sampled particulate matter during winter time in a rural region in California, where residential heating was done mainly by wood burning. From their
samples, which they collected during evening and nighttime, they derived $\alpha_{abs}$ values between 3.0 and 7.4. Sandradevi et al. (2008) investigated the spectral aerosol absorption in a small village in Switzerland, where the majority of the households also used wood burning for heating in winter. For this village, they found values for $\alpha_{abs}$ of 1.8-1.9. Nakayama et al. (2014) showed for the city of Nagoya, Japan, temporal variations for $\alpha_{abs}$ between summer
and winter of 1.21 and 1.43, respectively. For measurements in the Amazon basin during the dry season, Rizzo et al. (2011) reported $\alpha_{abs}$ values between 1.5 and 2.5 resulting from light-absorbing organic carbon released from biomass burning.

Despite their significance, the light absorption properties of carbonaceous aerosol are poorly investigated, mostly due to lack of appropriate instrumentation (Schmid et al. 2006).
Considering the organic mass fraction, which is the dominating component of carbonaceous aerosol in the atmosphere, even a small contribution of organic matter to light absorption may result in a significant absorption of solar radiation (Kirchstetter & Thatcher 2012, Hoffer et al. 2006).

For in situ absorption measurements, different kinds of instruments are used. Most in situ
measurements are performed with filter-based methods, where the aerosol particles are deposited on a filter. This works well for strongly absorbing aerosols. For aerosols with higher organic content and, therefore, with stronger scattering properties, filter-based absorption methods raise experimental uncertainties (Lack et al. 2008, Schnaiter et al. 2005, Weingartner et al. 2003, Bond et al. 1999, Hitzenberger 1993).

The deposition of the aerosol on a filter may change the physical properties of the particles and the combined optical properties of the deposit and filter substrate (Lack et al. 2008). Additionally, filter based methods require various corrections for multiple scattering on the filter substrate, for backscattering from the deposited particles and for the reduction of scattering with increasing particle load (Chow et al. 2009). This leads to large uncertainties
especially for atmospheric measurements where the aerosol optical properties are dominated by scattering. Under these conditions, an accurate absorption measurement with filter-based methods is limited by the strong cross-sensitivity of the method to the aerosol light scattering.

Unlike these filter based instruments, the photoacoustic method directly determines the light
absorption of aerosol particles in the airborne state. Due to the fact that the photoacoustic method is specific to the absorption properties of the particles there is no or only weak cross-sensitivity to particle light scattering, which makes this method ideally suited for atmospheric measurements (Rosencwaig 1975, Miklós et al. 2001)



In this study, we present a novel-single cavity photoacoustic spectrometer that measures the absorption coefficient of atmospheric aerosols at three wavelengths in the near-UV and visible spectral range. In section 2, we describe the instrument setup and discuss the instrument characterization in terms of laser power stability, linearity of the electronic setup, and detection limit together with calibration results. A laboratory study is presented in section 3, where combustion emissions from a propane flame were used as surrogates for atmospheric carbonaceous particles with different organic content. Finally, the instrument was applied in an urban field study in Karlsruhe, Germany. The wavelength-dependent absorption properties measured in this study are presented in Sect. 4. The laboratory and field studies have shown a reliable application of the novel photoacoustic spectrometer for ambient aerosol measurements.

## 2. INSTRUMENT SETUP

The absorption measurement by the photoacoustic method is initiated by a modulated laser. Due to the absorption of light by the particles within the laser beam, the carbonaceous material gets energetically excited and, consequently, heats up. The deposited thermal energy is subsequently released into the surrounding gas. Due to the periodic thermal expansion caused by the laser modulation, a pressure or sound wave is generated in the gas. This sound wave is amplified in an acoustic resonator and is detected by a microphone. The acoustic resonator and the microphone form the so-called photoacoustic cell (Rosencwaig 1975). The photoacoustic signal, PA, generated by the photoacoustic cell can be expressed as follows:

$$PA = b_{abs} \times P_{laser} \times c_{cell}. \qquad (1)$$

According to Eq. (1), PA [V] at the resonance frequency of the cell is determined by the aerosol light absorption coefficient $b_{abs}$ [m$^{-1}$], the laser power $P_{laser}$ [W] and a cell constant $c_{cell}$ [V W$^{-1}$ m].

The cell constant $c_{cell}$ is a function of the overall geometric setup of the laser, the microphone, and the resonance cell (Miklós et al. 2001). The value of $c_{cell}$ must be determined by calibrating the system with a material of known absorption cross section.

The setup of the newly developed photoacoustic instrument is shown in Figure 1. Multi-wavelength light absorption and heating of the sample in the photoacoustic cell is induced by three successively irradiating laser beams. Three diode laser modules (Qioptiq Nano 250 series), emitting at 445 nm, 532 nm, and 660 nm, are tuned to the resonance frequency of the cell. During operation, the laser powers are 150 mW, 100 mW, and 50 mW, respectively. The three laser beams are merged to the optical axis of the photoacoustic cell by beam splitters (DLMP 425, DLMP 505, Thorlabs). The laser power is continuously monitored by a power meter (PowerMax, Coherent) at the exit of the resonance cell. The photoacoustic cell was manufactured by HiLase Ltd., Hungary, according to our specifications and was modified by replacing the original windows by windows with specific broad band anti-reflection coatings. A resonance frequency of 1930 Hz was determined for the cell at a temperature of 25°C. The actual acoustic resonator is a pipe with open ends, having a length of 90 mm, a diameter of 8.5 mm, and a volume of 5.1 cm$^3$. The length of the pipe corresponds to λ/2 of the resonance frequency of the cell. With these geometric specifications, the pipe works as



an open acoustic resonator for the generation of longitudinal acoustic modes. Acoustic buffers are implemented on both sides of the resonator, each having a length of $\lambda/4$ and a diameter of 60 mm resulting in a buffer volume of about 130 cm$^3$. The length of the buffers and the sharp enlargement of the diameter between the resonator pipe and the buffer volume ensure (i) optimal back reflection of sound from inside the acoustic resonator, and (ii) resonance frequency filtering of noise from outside the resonator by destructive interference in the buffers. The buffers are terminated by optical windows, which seal the cell against the surrounding gas. Windows with broadband anti-reflection coatings are used to minimize residual reflections of the laser beam when entering and exiting the resonance cell.

The aerosol enters and exits the cell through the buffers which results in a 90° bend in the aerosol flow at the transition points from the buffer volumes to the acoustic resonator. A continuous aerosol flow of 1 std. liter per minute through the cell is maintained. Additionally, $\lambda/4$ notch filters are installed in line of the aerosol tube, to prevent flow noise from entering the resonator.

The microphone is connected to the resonator pipe through a small hole of 5 mm diameter close to the amplitude node of the fundamental acoustic mode of the cell, i.e., at the center of the pipe. In this way, the photoacoustically generated wave is detected with high sensitivity by a commercial hearing aid microphone (EK-3029, Knowles). This specific microphone was chosen for effective signal detection at resonance frequencies above 1000 Hz.

The microphone signal is fed to a lock-in amplifier (LIA-BVD-150-L, Femto). This lock-in amplifier is a two phase instrument, for phase sensitive detection, which provides signal X, magnitude R, and phase-shift Y with respect to the phase of the incident laser modulation frequency. Suitable-lock in parameters for the amplification and filter characteristics can be set for the input and output amplifier. The signal to noise ratio can be optimized by an additional low pass filter with adjustable time constant.

To determine the absorption by the aerosol particles, periodic background measurements are performed by passing the aerosol through a particle filter, which effectively removes the particles. The complete exchange of the gas in the cell volume between the background and aerosol measurements defines the minimum time that is required for the absorption measurement.

From equation (1) it is obvious that for accurate measurements a reliable control of the laser power during electronic recording is essential. The laser power stability for an operation period of 8 hours at room temperature was tested and showed maximum deviations from the specified power of less than 2% for all lasers. In detail, at 445 nm a laser power stability of 152 mW ± 0.2 mW, at 532 nm of 105 mW ± 0.4 mW, and at 660 nm of 52 mW ± 0.6 mW was found over this operation period. The linearity of the electronic detection and amplification system was checked by gradually decreasing the laser power while the cell was continuously flushed with nitrogen dioxide ($NO_2$) gas (10 ppm, Air Liquide). Figure 2 shows the result for the 445 nm, 532 nm, and 660 nm wavelengths. The tested power range was up to 150 mW for 445 nm, 100 mW for 532 nm, and 50 mW for 660 nm. The power measurements for the three lasers exhibit different slopes for each wavelength. This is attributeable to the wavelength-dependent absorption coefficient of $NO_2$ resulting in specific photoacoustic signals, which increase from 660 nm to 445 nm as shown in Fig. 2. Furthermore, the



photoacoustic response behavior shows high linearity with laser power for all three laser wavelengths. However, variations in the laser power during long term measurements can directly affect the photoacoustic signal. Those errors are reduced by simultaneously measuring the photoacoustic signal and the laser power.

The instrument setup has to be calibrated by a substance of known absorption cross section to determine the correlation between a given absorption coefficient and the measured photoacoustic signal in order to specify the cell constant of the setup (cf. Eq. 1). For this purpose, light absorbing nitrogen dioxide ($NO_2$) was used as calibration gas. Absorption cross sections for nitrogen dioxide are published by Voigt et al. (2002) over a broad

wavelength range and for 293 K and 1000 mbar temperature and total pressure, respectively. The derived absorption cross sections of $NO_2$ are 1.65e-6 $m^{-1}$ $ppb^{-1}$ for 445 nm, 3.56E-7 $m^{-1}$ $ppb^{-1}$ for 532 nm and 1.75E-8 $m^{-1}$ $ppb^{-1}$ for 660 nm. The measuring range of the lock-in amplifier can be regulated by adjustment of the gain settings. To determine cell constants at different gains, certified gas mixtures of $NO_2$ in synthetic air (Crystal-Mix, Air Liquide) of 5

ppm, 1 ppm, or 0.2 ppm were used. Based on the certified gas mixtures, different $NO_2$ gas concentrations are generated by diluting and mixing $NO_2$ standards with synthetic air. For each gain setting a distinct cell constant is determined.

In this study, the cell constants for the amplifier gain settings, i.e., lock-in sensitivities, 6 and 7 were determined. In Figure 3, the cell constant for the lock-in sensitivity 6 of 78518 ± 1123 V

$W^{-1}$ m is derived from calibration measurements at 532 nm (green squares) and 445 nm (blue squares). The concentrations of the deployed $NO_2$ calibration gas mixtures, indicated at the top axis of Figure 3, vary from 80 ppb up to 1000 ppb $NO_2$. On the bottom axis the corresponding absorption coefficients for $NO_2$ are plotted, calculated from the absorption cross sections for $NO_2$ (Voigt et al., 2002). The cell constant is then calculated from the

correlation slope between photoacoustic signal, laser power, and absorption coefficient according to Eq. 1. For the calibration at 532 nm, about one third of the measurements were originally done at sensitivity 7 and were subsequently rescaled to sensitivity 6 by applying the gain specifications given by the manufacturer.

We compared our novel three-wavelength instrument with a field-proven 532 nm

photoacoustic instrument of the Max Planck Institute for Chemistry, Mainz, Germany, which had been originally developed by Desert Research Institute, Reno, USA (Arnott et al. 1999). In Figure 4, absorption coefficients at 532 nm measured with both instruments are compared for an aerosol chamber experiment with BC aerosol from the Combustion Aerosol Standard (CAST) generator (see Sec. 3 for details on the CAST generator and the chamber

experiments).

During the experiment both instruments measured absorption coefficients between $2\times10^{-5}$ $m^{-1}$ to $1.7\times10^{-4}$ $m^{-1}$. The slope of the regression line is 1.03 ± 0.02 and the coefficient of determination, $R^2$=0.98, indicates good agreement of both instruments at 532 nm over the measured range. Given that two different instruments with different laser and detection

systems were compared, this is a very convincing result.

The limit of detection (LOD) of the photoacoustic instrument was determined from four different days of field measurements by analyzing 10 background (particle free ambient air)



measurements for each wavelength. The LOD was then calculated according to the German Standard DIN 32645

$$LOD = \overline{b_a^{BG}} + 3 \times \sigma^{BG} \tag{2}$$

with $b^{BG}$ the mean background absorption coefficient and $\sigma^{BG}$ the corresponding standard deviation. LOD values of $5.6 \times 10^{-6}$ m$^{-1}$, $6.6 \times 10^{-6}$ m$^{-1}$, and $1.8 \times 10^{-5}$ m$^{-1}$ were deduced for the wavelengths 445 nm, 532 nm, and 660 nm, respectively.

# 3. LABORATORY STUDIES ON CAST SOOT OF DIFFERENT C/O FUEL RATIOS

Two sets of experiments were performed at the aerosol and cloud chamber facility AIDA. One set was conducted during the SOOT11 campaign in 2010. The second set of experiments was done during the SOOT15 campaign in June 2013. While the SOOT11 campaign took place at the 84 m$^3$ sized AIDA chamber (Wagner et al. 2009) SOOT15 was performed at the smaller stainless steel chamber NAUA with a volume of 3.7 m$^3$ (Schnaiter et al. 2006, Linke et al. 2006).

Before each experiment, the chamber was evacuated, flushed and refilled with particle-free synthetic air. Combustion Aerosol Standard (CAST) aerosol, generated by a co-flow diffusion burner (miniCAST Series 5200, Jing Ltd., Switzerland), was filled into the clean chamber. The miniCAST generator is a compact version of the first instrument series that had been used in the soot absorption study by Schnaiter et al. (2006). Further details on the operation of the burner together with a characterization of the particle emission can be found there. Note that due to mechanical design changes, the composition characteristics of the emitted particles found by Schnaiter et al. (2006) has changed for the miniCAST version. After the aerosol addition to the NAUA chamber, the aerosol was continuously stirred by a mixing fan to ensure homogeneous gas and particle conditions throughout the chamber volume. Stainless steel tubes of 6 mm diameter were used for aerosol sampling. The aerosol flow through the connected photoacoustic instruments, an integrating nephelometer (TSI, model 3563), and a condensation particle counter (CPC; TSI, model 3022A) was 1 L min$^{-1}$, 5 L min$^{-1}$, and 1.5 L min$^{-1}$, respectively.

The miniCAST burner was operated with a mixture of propane and synthetic air. Different mass ratios of fuel to air were adjusted to operate the burner at different burning conditions. In this way, the ratio of elemental carbon (EC) to organic carbon (OC) of the generated carbonaceous aerosol could be altered over a broad range (Schnaiter et al. 2006). At low C/O ratios the fraction of EC is dominating in the aerosol generated by the burner. When the C/O ratios increase, the fraction of EC is reduced while the OC content of the generated aerosol increases. In our studies the EC content was varied between 10% and 50% of the total carbon (TC) mass.

The TC, EC, and OC contents of the aerosol samples were determined from quartz fiber filters analyzed off-line by a Sunset OC/EC thermal analyzer (Sunset Laboratory Inc., USA)



using the EUSAAR-2 temperature protocol (Cavalli et al. 2010). During the thermal-optical measurement, the filter medium is first heated stepwise to 600°C in an inert helium atmosphere to desorb OC, then cooled to 500°C, and reheated in several temperature steps to 800°C under a helium/oxygen atmosphere to oxidize EC. To correct for pyrolysis of OC in

the inert atmosphere, which would lead to a positive EC artefact, a laser signal is used as optical control to determine the OC-EC split.

The mass concentration of refractory black carbon (rBC) in the carbonaceous aerosol was determined by a single particle soot photometer SP-2 (DMT, USA). The SP-2 utilizes incandescence of single rBC particles in a laser cavity to quantify the particle

mass down to the sub-fg level. The incandescence signal of the instrument was calibrated with size selected fullerene soot particles (Alfa Aesar). For analysis of the SP-2 data we used the software toolkit developed by Martin Gysel from the Paul Scherrer Institute, Switzerland (http://aerosolsoftware.web.psi.ch/). Details on the instrument characteristics, accuracy, and reproducibility as well as the operation at the AIDA facility were described by Laborde et al.

15    (2012).

## Photoacoustic measurements during the SOOT11 campaign

During the SOOT11 campaign, miniCAST experiments with two different C/O fuel ratios were performed. One experiment was done with a C/O ratio of 0.29, which corresponds to an OC/TC ratio of about 60% and the second experiment was conducted at a C/O ratio of 0.4

corresponding to an OC fraction of about 90%. At the beginning and the end of each experiment, filter samples were taken for off-line OC/EC analysis. Details of the instrumental setup are given by Laborde et al. (2012). In SOOT11, only the single wavelength photoacoustic instrument from the Max Planck Institute for Chemistry was available, which measured the absorption coefficient at 532 nm.

Mass-specific absorption cross sections (MAC) at 532 nm were deduced from the photoacoustic absorption coefficients in conjunction with the EC values obtained from the thermo-optical analysis. The MAC (532) found for C/O ratios of 0.29 and 0.4 were $14.5\pm1.6$ $m^2\,g^{-1}$ and $18.6\pm 4.2\ m^2\,g^{-1}$, respectively.

Alternatively, the MAC (532) can be determined by using the rBC mass concentrations

measured by the SP-2 instrument resulting in a MAC (532) for the C/O ratio 0.29 of 12.5 $m^2$ $g^{-1}$ (Laborde et al. 2012). It was not possible to determine the corresponding MAC (532) for the C/O ratio of 0.4 in the same way as the rBC concentration during this experiment was too low and the SP-2 did not provide incandescence signals for this aerosol type (Gysel et al. 2012).

## Photoacoustic measurements during the SOOT15 campaign

During the SOOT15 campaign, miniCAST soot experiments at four different burning conditions were conducted. The corresponding C/O ratios of 0.25, 0.29, 0.33 and 0.38 resulted in OC/TC ratios of the combustion aerosols between 50% and 90% (Table 1). At the





beginning of each experiment, a filter sample was taken for off-line OC/EC analysis. The experimental setup is shown in Figure 5.

Particle number concentrations were measured (i) directly from the chamber and (ii) after a dilution stage (PALAS, 3xVKL10). High particle number concentrations in the chamber caused strong coagulation of the aerosol during the first part of the experiment. Initial particle
number concentrations in the chamber varied between $3\times10^4$ and $8\times10^4$ $cm^{-3}$.

The particle size distribution was measured with a scanning mobility particle spectrometer (SMPS), composed of a differential mobility analyzer (DMA 3071, TSI) and a condensation particle counter (CPC 3010, TSI). Due to the aerosol preparation and coagulation in the
chamber, the median particle sizes measured in these experiments ranged from 80 nm to 250 nm.

For the optical characterization of the chamber aerosol, the absorption and scattering coefficients were determined with our new three-wavelength photoacoustic spectrometer in combination with a three-wavelength integrating nephelometer (model 3563, TSI). Before the
experiments, the nephelometer was calibrated with $CO_2$ and air. Details on nephelometer operation at the AIDA facility, including a discussion of the error corrections, are given by Schnaiter et al. (2005). Both instruments were connected directly to the aerosol chamber. The scattering coefficients were measured continuously. An automated pneumatic valve (Swagelok, Germany) was used upstream of the inlet of the photoacoustic instrument to
change between particle and particle-free background measurements in periodic cycles of about 7 minutes.

## Optical properties of CAST soot

From the photoacoustic and integrating nephelometer measurements, spectrally resolved absorption and scattering coefficients $b_{abs}(\lambda)$ and $b_{sca}(\lambda)$ were determined and the following
optical parameters were derived

(i)   spectrally resolved mass-specific absorption cross sections, $MAC(\lambda)$
(ii)  Ångström exponents of absorption, $\alpha_{abs}$, and scattering, $\alpha_{sca}$
(iii) wavelength resolved single scattering albedo, $\omega(\lambda)$

These parameters are summarized in Table 1 for the different miniCAST aerosol types and
discussed in more detail in the following sections.

## Specific Absorption cross sections

In this study, time resolved mass-specific absorption cross sections $MAC(\lambda)$ were derived from the concurrent measurements of the photoacoustic absorption coefficients and the rBC mass from the SP-2. The filter sampling for EC/OC analysis could be done only at the
beginning of each experiment to determine the OC/TC fraction of the aerosol at each C/O ratio of the miniCAST. In order to avoid perturbation of the aerosol sampling during the optical measurements, no filter sampling was possible in parallel with the experiments,



therefore no comparisons of BC mass from SP-2 to EC mass from filter measurements are available for the SOOT15 experiments.

With increasing C/O fuel ratio in the flame of the miniCAST generator, the organic content of the emitted carbonaceous aerosol rises. The OC fraction in the total carbon aerosol mass at 5 C/O ratio of 0.25 is about 50% and increases towards higher C/O ratio up to 90% or more.

The color of the aerosol filter samples taken from the chamber already indicated a change in the optical behavior of the emitted aerosol. While the quartz filters were deep black at C/O ratios of 0.25 and 0.29, the filter sample at 0.38 was of brownish appearance (Figure 5).

The MAC at 660 nm, which was the longest wavelength measured here, remained almost 10 constant with increasing organic content of the aerosol (Table 1). In contrast, the MAC at 532 nm clearly increased with increasing C/O ratio from 10.4 ±2.3 to 21.0 ±2.5 $m^2$ $g^{-1}$. An even stronger functional dependence was observed for the MAC determined for the shortest investigated wavelength of 445 nm. For this wavelength, the MAC increased from 12.5 ±1.7 $m^2$ $g^{-1}$ at a C/O ratio of 0.25 to 31.0 ±5.9 $m^2$ $g^{-1}$ at a C/O ratio of 0.38. Here, the MAC(445 15 nm) of the carbonaceous aerosol with a OC content of about 90% was 2.5 times higher than the corresponding MAC value of the aerosol with an OC content of only 50%. As the MAC in this case is based on the SP-2 rBC mass measurement, this result clearly shows that the OC material that was co-emitted with the BC must have a non-negligible absorption cross section with a strong wavelength dependence.

20 The results of the SOOT15 campaign were in agreement with the results of the SOOT11 study, where the absorption coefficients as well as the carbonaceous content of the samples were determined with partly different instrumentation at C/O ratios of 0.29 and 0.4, but only at 532 nm.

In SOOT15, the determined value of the MAC (532) at a C/O ratio of 0.29 was 16.4 ±0.4 $m^2$/g 25 (Table 1). This fits quite well to measurements done during the SOOT11 campaign, where we determined a MAC (532) of 14.5 ±1.6 $m^2$ $g^{-1}$ with the 1 $\lambda$-PAS (DRI) and thermo-optical EC. When comparing the MAC (532) value for C/O 0.29 of SOOT15 with the corresponding MAC (532) value of 12.5 $m^2$ $g^{-1}$ from SOOT11 that was based on the SP-2 incandescence measurement (Laborde et al. 2012), both values agree within a range of about 30%. Due to 30 the high organic carbon content of around 90%, the CAST aerosol produced at a C/O ratio of 0.38 in SOOT15 should be comparable to the corresponding aerosol in SOOT11, which was produced at a C/O ratio of 0.4. For SOOT15 we derived a MAC (532) value of 21.0±2.5 $m^2$ $g^{-1}$ for this aerosol type. For SOOT11 the corresponding value was 18.6 ±4.2 $m^2$ $g^{-1}$. Both values again agree nicely, given the fact that they were measured with different instruments. 35 A reliable SP-2 incandescence measurement at these high C/O ratios was found to be impossible and for this reason no rBC mass specific absorption cross section could be specified.

## Ångström exponents

The Ångström exponents of the absorption and scattering coefficients were deduced by 40 fitting a power law function to the three measured coefficients, respectively:



$$\log b_{abs,sca}(\lambda 2) = \alpha_{abs,sca} * \log(\lambda 2/\lambda 1) + \log b_{abs,sca}(\lambda 1). \qquad (3)$$

Carbonaceous material with $\alpha_{abs}$ of unity only shows a slight spectral behavior as described for BC in the UV-VIS spectral wavelength range by Schnaiter et al. (2003) and Moosmüller (2009). For the miniCAST burner, a fuel-to-air ratio with C/O of 0.29 is close to the conditions for the stoichiometric combustion. For this C/O ratio, we derived the lowest values for the Ångström exponents $\alpha_{abs}$ and $\alpha_{sca}$ of 1.3 ±0.2 and 2.2 ±0.02, respectively (Table 1). The $\alpha_{abs}$ value of 1.3 is rather low and represents a weak wavelength dependence. With increasing organic content in the soot samples, both Ångström exponents clearly increase, reflecting a steeper wavelength dependence of both coefficients, $b_{abs}(\lambda)$ and $b_{sca}(\lambda)$. At a C/O ratio of 0.38 with an organic content of almost 90%, the Ångström exponent $\alpha_{abs}$ reached 3.1 ±1.0 while $\alpha_{sca}$ increased to 3.4 ±0.1.

## Single scattering albedo

The single scattering albedo $\omega$ is defined as the ratio of the light scattering coefficient to the extinction coefficient, i.e. the sum of the scattering and absorption coefficients. $\omega(\lambda_s)$ was deduced from the measured scattering and absorption coefficients at the wavelength positions $\lambda_s$ of the integrating nephelometer. For that, the absorption Ångström exponents $\alpha_{abs}$ of Table 1 were used to inter- and extrapolate the photoacoustic absorption coefficients measured at 445 nm, 532 nm and 660 nm to absorption coefficients at the nephelometer wavelengths, i.e. at 450 nm, 550 nm, and 700 nm.

After adjusting the absorption coefficients to the corresponding nephelometer wavelengths the single scattering albedo of the aerosol at 450 nm, 550 nm and 700 nm was calculated and is presented in Table 1.

The single scattering albedo $\omega(\lambda)$ increases with increasing C/O ratio of the carbonaceous aerosol.

## Discussion of the chamber results

In the presented chamber experiments, we determined MAC of combustion aerosols with increasing organic content. The rBC mass measured by the SP-2 incandescence method was compared to the off-line elemental carbon (EC) and total carbon (TC) analysis results that were obtained by the thermo-optical method. The experiments show that for aerosols with higher organic content the MAC of rBC (MAC-rBC) and EC (MAC-EC) increases.

On the other hand, due to the increase in the OC mass, the MAC of TC (MAC-TC) decreases with increasing C/O ratio. Schnaiter et al (2006) found MAC-TC with the predecessor version of the CAST burner of 5.5 ±0.7 $m^2$ $g^{-1}$ and 3.8 ±0.5 $m^2$ $g^{-1}$ for combustion aerosol produced with a C/O ratio of 0.29 and 0.4, respectively. Relating the absorption coefficients measured at 532 nm during SOOT11 to the TC mass concentration, MAC-TC values of 5.4 ±0.8 $m^2$ $g^{-1}$ and 1.6 ±0.2 $m^2$ $g^{-1}$ are deduced for the C/O 0.29 and C/O 0.4 combustion aerosol, respectively. Comparing the MAC from both studies, the values nicely agree for the C/O ratio of 0.29 but differ significantly for the C/O ratio of 0.4. The latter discrepancy is due to the fact



that different CAST burner models were used. Schnaiter et al. (2006) used the predecessor version of the CAST burner, which significantly differs in the dimensions of the combustion chamber compared to the miniCast 5200 burner that was used in the SOOT11 and SOOT15 studies. Due to these differences, the flaming conditions in both burner models are different resulting in different EC/OC versus C/O characteristics for the emitted combustion aerosol (see a comparison of both burners in Crawford et al. (2011)), which limits direct comparisons of the EC/OC ratios of the aerosol emitted for similar C/O ratios. As shown by Crawford et al. (2011), combustion at a C/O ratio of 0.4 in the miniCast model produces aerosol with an EC/OC ratio that corresponds to the aerosol emission for flaming conditions at C/O ratios around 0.8 in the predecessor version of the CAST burner. Note that the thermographic analysis of EC, TC, and OC in Schaiter et al. (2006) was done according to the VDI-method (Ulrich et al., 1990).

In a laboratory study by Kirchstetter and Novakov (2007), the MAC values of BC aerosol generated with a diffusion flame were determined. The generated BC particles, which were analyzed by a thermal optical analysis (TOA) method described by Novakov (1981), contained no significant amounts of OC. Kirchstetter and Novac used a particle soot absorption photometer (PSAP) to measure the absorption coefficient at a wavelength of 530 nm. This measurement was then related to the "BC" mass of the TOA method, or in this case, because OC was negligible, the EC mass, to deduce a MAC of 8.5 $m^2$ $g^{-1}$ for the combustion aerosol. A single scattering albedo, $\omega(550\ nm)$, of 0.15 was determined from the PSAP data and simultaneous measurements of the scattering coefficient with an single wavelength integrating nephelometer. In our study, the BC aerosol with the lowest OC content of 50% (C/O=0.25), resulted in a MAC-rBC at 532 nm of 10.4 ±2.3 $m^2$ $g^{-1}$ and a $\omega$ at 532 nm of 0.12. Comparing both results it can be concluded that for 532 nm even a significant increase in the OC content of the aerosol only results in a moderate increase of the MAC. This finding relates to results obtained in field measurements by Kondo et al. (2009), who analyzed ambient aerosols at six rural and urban sites in Asia, which are strongly impacted by vehicle and/or biomass burning emissions. For their measurements they used two different filter-based photometers, a Particle Soot Absorption Photometer (PSAP) and a Continuous Soot Monitoring System (COSMOS). To remove the volatile aerosol components they used a heated inlet system at temperatures of 400°C before measuring BC on the filter. Together with mass measurements (EC/OC analyzer, Sunset Laboratory) they found for samples of widely different BC sources MAC values at 565 nm of 10.5 ±0.7 $m^2$ $g^{-1}$ for the remaining BC.

The light absorption by organic species increases towards shorter wavelengths. Lewis et al. (2008) investigated the optical properties of different biomass burning aerosols during laboratory measurements in the Fire Laboratory Missoula Experiment (FLAME). For different biomass types, they determined the Ångström exponent, $\alpha_{abs}$, in the wavelength range covered by their measurements (405 nm, 532 nm and 870 nm). From these experiments, they found that for several biomass fuels the span of Ångström exponents is greater in the wavelength range between 405 nm and 870 nm than between 532 nm and 870 nm and concluded that the particle emissions from these fuels absorb near-UV radiation much more efficiently. Similarly, in our measurements the $\alpha_{abs}$, determined between 445 nm and 660 nm, increased significantly with increasing organic content of the aerosol (Tab. 1). While the carbonaceous aerosol with an organic content of about 60% has a moderate $\alpha_{abs}$ of 1.3, the





carbonaceous aerosol with almost 90% organics reaches $\alpha_{abs}$ = 3.1. This increase reflects the stronger MAC increase at shorter wavelengths in case of combustion aerosol with a high organic content (Tab. 1).

To estimate the contribution of brC to the total absorption at 445 nm for the different C/O fuel ratios, we calculated the hypothetical fraction of absorption by BC at 445 nm assuming (i) a $\alpha_{abs}$ of 1.0 for BC in all four experiments listed in Table 1 and (ii) the MAC-rBC at 660 nm to be due only to BC. From the difference between the measured MAC-rBC at 445 nm and MAC-rBC at 445 nm calculated only for the hypothetical BC fraction, the fraction of absorption by brC in the light absorbing mass can be estimated. The fraction of absorption by brC during these experiments changes from 5.1% at C/O 0.25 to 55% at C/O 0.38.

## 4. URBAN FIELD STUDY

The multi-wavelength photoacoustic spectrometer was operated for the first time in the field during a campaign in October/November 2012. The measuring site at Durlacher Tor is a central traffic junction in the urban area of Karlsruhe, Germany. This site is generally characterized by high traffic volume, but at the time of the campaign, intense construction work for a new subway tunnel was taking place.

The measuring site was located next to a three-lane road and close to a street crossing. The aerosol was sampled at a distance of 5 m from the road and a height of 3 m above ground using a PM2.5 head (DPM2.5/1/00, Fa.Digitel) with a sampling volume of 1 m$^3$ h$^{-1}$. The sampled aerosol was led down to the instruments, which were located beneath the sampling head inside a simple cabin with a connection for power supply. The aerosol inlet flow enters the cabin via a 14 mm diameter stainless steel tubing. Within a flow splitter, 5 L min$^{-1}$ and 2.6 L min$^{-1}$ were taken isokinetically from the aerosol inlet flow. From the splitter, 5 L min$^{-1}$ were led through the nephelometer and 2.6 L min$^{-1}$ through the joint flow line for the 3-λ-photoacoustic spectrometer with a sample flow of 1 L min$^{-1}$, the SP-2 with a sample flow of 0.12 L min$^{-1}$ and the CPC (model 3775, TSI) with a sample flow of 1.5 L min$^{-1}$. Upstream of the 3-λ-photoacoustic spectrometer a diffusion dryer (filled with silica gel) was installed. The setup of the instruments during the campaign is shown in Figure 6.

Additionally, filter samples were taken with a second sampler unit using a PM2.5 head and a sampling volume of 2.3 m$^3$ h$^{-1}$ on quartz filters for off line thermo-optical EC/OC analysis. Each day the filter sampling took place for 23.5 h from 14:00 to 13:30 local time of the next day.

The campaign started on 16 October and ended on 6 November 2012 . During the first two weeks the weather was sunny with daytime temperatures around 15°C and without any rain. On 27 October, the weather changed considerably from warm and sunny autumn weather to cold nights and cold but sunny days, which initiated also the start of the heating season.

## Results from the urban field campaign

Here, we exemplarily show and discuss the measurement results for 31 October. In Figure 7, the SP-2 mass measurements are shown (black line; left axis) as refractory rBC mass





concentration. Furthermore, the absorption coefficients from the photoacoustic measurements are shown for the three instrument wavelengths (blue, green and red squares; right axis). The figure shows a good correlation between rBC mass concentration and the absorption coefficients. Typical diurnal variations in rBC mass concentration due to

rush hour traffic in the morning and traffic and domestic heating in the evening can be observed.

Figure 8 presents the particle number concentration of scattering particles deduced from the SP-2 measurements (left axis). While the grey line represents the number concentration of all scattering particles, the light blue line only shows the number concentration of those

scattering particles that have no incandescence signal, i.e., contain no rBC mass. The scattering coefficients obtained from the nephelometer are shown for the three nephelometer wavelengths (red, green, and blue; right axis). The trend of the nephelometer data matches the measured number concentration of all scattering particles (grey line) while there is no correlation with the number concentration of rBC-free scattering particles. This clearly

indicates that the light scattering coefficient was dominated by particles that are linked to combustion processes (traffic or heating emissions).

We derived MAC-rBC($\lambda$) values from the photoacoustic absorption coefficients and SP-2 mass measurements. These values were averaged over a time period of 24 h, starting at midnight. The absorption coefficients were derived with a time resolution of almost 14 min.

The time resolution of the SP-2 incandescence measurement was matched to the time resolution of the photoacoustic measurement. The absorption coefficients were then related to the associated SP-2 rBC mass concentration in the corresponding time window. Over one 24 h cycle, about 100 individual values for MAC-rBC($\lambda$) were determined. As shown in Table 2, we obtained a MAC-rBC(660) value of 7.5 ±4.9 $m^2$ $g^{-1}$, a MAC-rBC(532) value of

8.4 ±3.1 $m^2$ $g^{-1}$ and a MAC-rBC(445) value of 12.9±2.8 $m^2$ $g^{-1}$.

These data can be compared to the MAC-EC($\lambda$) values derived by relating the photoacoustic measurement to the EC mass concentration determined by the thermo-optical method. For this, the absorption coefficients were averaged over the 24 h time period of the filter sampling and then related to the EC mass concentration obtained from off-line filter analysis. We

obtained a MAC-EC(660) value of 6.9 ±2.7 $m^2$ $g^{-1}$, a MAC-EC(532) value of 7.7±3.5 $m^2$ $g^{-1}$ and a MAC-EC(445) value of 11.6 ±6.2 $m^2$ $g^{-1}$. The MAC($\lambda$) values determined from the SP-2 incandescence and the Sunset thermo-optical analysis are in very good agreement keeping in mind that two different techniques were used to obtain the rBC and EC mass concentrations (Table 2).

To provide evidence that residential wood burning in the evening hours increases the OC fraction in the carbonaceous aerosol and, consequently, increases the shortwave MAC at this urban site, we compared average MAC values for two time windows. For the traffic-dominated period, we chose the time between 07:00. to 09:00 local time For the period, which was likely to be influenced by residential wood burning, we selected the time between

20:00 to 22:00. The MAC values for the traffic-dominated period we found MAC-rBC(445)=10.8±1.0 $m^2$ $g^{-1}$, MAC-rBC(532)=7.0±0.8 $m^2$ $g^{-1}$ and MAC-rBC(660)=5.9±1.5 $m^2$ $g^{-1}$. The averaged MAC values for evening hours were MAC-rBC(445)=14.8±4.2 $m^2$ $g^{-1}$, MAC-rBC(532)=8.6±2.0 $m^2$ $g^{-1}$ and MAC-rBC(660)=6.0 ±1.7 $m^2$ $g^{-1}$, thus showing a clear increase in MAC in the blue spectral range during the period influenced by residential wood burning.


The Ångström exponents of absorption derived for 31 October are shown in Table 3. Averaging the measurements over the 24 h period results in an $\alpha_{abs}$ value of 1.9 for the 445 nm to 660 nm spectral range. When $\alpha_{abs}$ was specifically calculated for the shortwave range between 445 nm and 532 nm and the longwave range between 532 nm and 660 nm, values
for $\alpha_{abs}(445-532)$ and $\alpha_{abs}(532-660)$ of 2.6 and 1.3 were derived, respectively.

In Table 4, the single scattering albedo, $\omega(\lambda)$, at the three nephelometer wavelengths and the scattering Ångström exponent $\alpha_{sca}$ is given for the measurements at Durlacher Tor. The values for $\omega(\lambda)$ are in the range typical for atmospheric aerosols, but are significantly higher than the values given in Table 1 for the chamber experiments. The value of the scattering
Ångström exponent, $\alpha_{sca}$, is 1.7±0.2, and is significantly lower than the corresponding value deduced for all four chamber experiments (Table 1). A possible explanation for these observations is that in contrast to the laboratory-generated combustion aerosol, the aerosol at Durlacher Tor is likely burdened with additional dust particles such as released by the large construction area.

## Comparison of the field results with literature data

The MAC determined for freshly generated BC at 550 nm is in the range of 7.5 ±1.2 $m^2\ g^{-1}$ (Bond et al. 2013, Bond & Bergström 2006). These authors state that the MAC(550) will increase by up to 50% when the BC aerosol is internally mixed with other aerosol compounds. For brown carbon, a weak light absorption with a MAC(550) of about 1 $m^2\ g^{-1}$ is
usually given in the literature (Bond et al.2013).

From field studies, a range of different MAC($\lambda$) was published (Kondo et al. 2009, Knox et al. 2009, Laborde et al. 2012a, Petzold et al. 2002, Snyder & Schauer 2007, Thompson et al 2012,). Additionally, different protocols for the EC/OC mass analysis hamper the comparison of MAC values given in the literature. Chan et al (2011) reported MAC(781) values for rural,
suburban and urban locations in Canada, which differ between 9 to 43 $m^2\ g^{-1}$ when using the NIOSH 5040 protocol and between 6 to 27 $m^2\ g^{-1}$ when using the IMPROVE protocol for the EC/OC mass analysis. In our study, we used the EUSAAR 2 protocol, which was set up to optimize charring corrections and to minimize biases in EC and split point determination (Cavalli et al. 2010).

At 532 nm we derived MAC(532) values of 8.4 $m^2\ g^{-1}$ and 7.7 $m^2\ g^{-1}$ when relating to SP-2 and filter-based mass measurements, respectively (Table 2). Interpolating these values to 550 nm (with $\alpha_{abs}(445-660)=1.9$) results in MAC(550) values of 7.9 $m^2\ g^{-1}$ and 7.2 $m^2\ g^{-1}$, which is in very good agreement with the average value of 7.5±1.2 $m^2\ g^{-1}$ derived by Bond and Bergström (2006) for freshly emitted BC.

Especially in the blue range of the visible spectrum there are fewer literature data available for the MAC. From biomass burning experiments during FLAME 3, McMeeking et al. (2014) reported an average MAC(405) of 9.8±1.5 $m^2\ g^{-1}$ determined from combined photoacoustic and single-particle mass spectrometric measurements for heated and unheated aerosol samples with minimal coating.



Healy et al. (2015) reported an averaged MAC(405) of 8.4 $m^2$ $g^{-1}$ for a traffic site in Toronto, Canada, from combined photoacoustic and single-particle mass spectrometric measurements. Ueda et al. (2015) measured soot containing aerosols at the NOTO site in Suzu City, Japan. They reported values of photoacoustic absorption coefficients and SP2

mass concentrations, which result in MAC(405) values between 11.1 $m^2$ $g^{-1}$ and 15.8 $m^2$ $g^{-1}$.

Olson et al. (2015) reported MAC data in the wavelength range between 880 nm and 370 nm from various source emissions measured by an aethalometer, a photoacoustic extinctiometer, and thermo-optical EC/OC mass analysis. Multi-wavelength absorption was measured for aerosol produced by different fuels including wood, agricultural biomass, coals,

plant matter, and petroleum distillates in controlled combustion settings. They reported bulk MAC (660/520/470/370) values for Diesel aerosol of 6.61/8.3/9.28/11.07 $m^2$ $g^{-1}$, for wood burning aerosol of 2.66/3.34/3.73/4.52 $m^2$ $g^{-1}$, and for pellets burning of 2.69/7.15/14/49.91 $m^2$ $g^{-1}$.

For our traffic-dominated, but residential heating influenced measurements, we found

MAC(445) between 11.6 $m^2$ $g^{-1}$ and 12.9 $m^2$ $g^{-1}$ (Table 2), which is in the range of the literature data for fresh traffic emissions. Calculating the difference between the determined MAC at 445 nm of the whole aerosol and the MAC(445) calculated only for BC using an $\alpha_{abs}$ of 1.0 accounts for the fraction of brC. In this case the calculated absorption fraction by brC is in the range between 11.8% and 13.8%, which indicates that the aerosol absorption was

dominated by traffic emissions.

During biomass burning experiments, McMeeking et al. (2014) and Liu et al. (2014) showed that $\alpha_{abs}$ is strongly related to $\omega(\lambda)$. McMeeking et al. found $\alpha_{abs}$ of 1.5 to 7 and $\omega(781)$ values of 0.4 to 1.0. For $\alpha_{abs}$ around 2 they found $\omega(781)$ below 0.8, while when $\alpha_{abs}$ rapidly increases up to 7, the $\omega(781)$ value approaches 1.0. These results are in good agreement

with our data given in Table 3 and Table 4. Chakrabarty et al. (2010) determined $\alpha_{abs}$ and $\omega(\lambda)$ of tar balls from agricultural biomass combustion of duff. The measurements result in $\alpha_{abs}(405\text{-}532)$ of 6.4 and $\omega(405)$ of 0.95, $\omega(532)$ of 0.98 and $\omega(780)$ of 0.98. As a possible criterion for identifying brC, Chakrabarty et al. (2010) proposed a negative Ångström exponent of $\omega(\lambda)$.

While Schnaiter et al. (2006) reported negative Ångström exponents of $\omega(\lambda)$ for CAST soot experiments at high OC content, neither during our current laboratory study nor at the field measurements negative Ångström exponents of $\omega(\lambda)$ were found.

## 5. CONCLUSION

For an effective characterization of the optical properties of carbonaceous aerosols the accurate determination of spectrally resolved absorption coefficients is essential. The discussion of light absorbing organic species makes a wavelength-dependent determination of the absorption coefficients indispensable. Ideal measurements should cover the whole wavelength range from the near UV to visible to the near IR.



During chamber and field experiments, our novel single-cavity photoacoustic spectrometer was found to accurately measure spectrally resolved absorption coefficients at three wavelengths. The advantage of a single cavity instrument is that the cell constant can be successfully determined at different wavelengths.

Different artificially generated carbonaceous aerosols with increasing organic content were investigated. From these measurements, increasing mass-specific absorption cross sections (MAC) were derived towards shorter visible wavelengths. It was shown that the shortwave MAC strongly increases with organic content of the aerosol. This sensitivity to the OC content of the combustion aerosol was also reflected by the single scattering albedo, $\omega(\lambda)$, and the
absorption Ångström exponent, $\alpha_{abs}$, which both increase towards higher organic content of the aerosol.

Comparing the laboratory deduced MAC values with the values derived for ambient carbonaceous aerosol measured at an urban field site it became obvious that the aerosol at the site was dominated by traffic emissions. However, a specific investigation of the diurnal
variations in the spectral absorption measurements showed that the aerosol was influenced by residential wood burning during the evening hours. These first field results with our photoacoustic spectrometer are encouraging and strengthen the necessity for long-term measurements of the spectral aerosol absorption in urban environments.

*Acknowledgements: This work was funded by the Helmholtz-Gemeinschaft Deutscher*
*Forschungszentren as part of the program "Atmosphere and Climate" and a Start-up budget of the "KIT Kompetenzbereich Erde und Umwelt". The SP-2 and the development of the PAS was funded by the HGF Ausbauinvestition ATMONSYS. We thank the AIDA team and especially Georg Scheurig, Thomasz Chudy and Steffen Vogt for their technical support. We would like to thank Reiner Gebhardt (IfGG, KIT) who installed the cabin and facilities at*
*Durlacher Tor. We gratefully acknowledge Otmar Schmid (formerly Max Planck Institute for Chemistry), Uli Pöschl (Max Planck Institute for Chemistry) and the Max Planck Society for their support.*

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



Table 1:   Optical properties resulting from different C/O ratios of mini-CAST soot

| C/O | OC/TC | SP-2 | σ Abs 445nm | σ Abs 532nm | σ Abs 660nm | ω(450) | ω(550) | ω(700) | αabs | αsca |
|------|-------|--------|-------------|-------------|-------------|-----------|-----------|-----------|---------|----------|
|      | %*    | µg m⁻³ | m² g⁻¹      | m² g⁻¹      | m² g⁻¹      |           |           |           |         |          |
| 0.25 | 50±2  | 16±5   | 12.5 ±1.7   | 10.4 ±2.3   | 8.0 ±3.4    | 0.18 ±0.05 | 0.12 ±0.03 | 0.12 ±0.06 | 1.7 ±0.5 | 2.3 ±0.06 |
| 0.29 | 62±8  | 34 ±1  | 14.1 ±0.9   | 16.4 ±0.4   | 9.1 ±0.8    | 0.19 ±0.01 | 0.13 ±0.01 | 0.13 ±0.01 | 1.3 ±0.2 | 2.2 ±0.02 |
| 0.33 | FF    | 29±18  | 19.0 ±2.9   | 17.7 ±3.0   | 9.9 ±3.1    | 0.32 ±0.03 | 0.21 ±0.04 | 0.21 ±0.03 | 1.5 ±0.2 | 2.3 ±0.1 |
| 0.38 | 89±2  | 3 ±0.4 | 31 ±5.9     | 21 ±2.5     | 9.4 ±2.9    | 0.38 ±0.04 | 0.19 ±0.04 | 0.27 ±0.07 | 3.1 ±1.0 | 3.4 ±0.1 |

\* filtersamples were taken before optical measurements

FF- incorrect filter sampling



Table 2: Specific absorption cross sections determined from photoacoustic, incandescence and thermo-optical measurements for 31 October 2013 (24h average) at Durlacher Tor, Karlsruhe

| $\lambda$ (nm) | 445 | 532 | 660 |
|---|---|---|---|
| $\sigma_{Abs}$ (m$^2$g$^{-1}$) (rBC SP-2) | 12.9 ±2.8 | 8.4 ±3.1 | 7.5 ±4.9 |
| $\sigma_{Abs}$ (m$^2$g$^{-1}$) (EC Sunset) | 11.6± 6.2 | 7.7 ±3.5 | 6.9 ±2.7 |



Table 3:  Angström exponents of absorption, $\alpha_{abs}$, derived
for 31 October 2013 (24h average) at
Durlacher Tor, Karlsruhe

| Range $\lambda_1$ to $\lambda_2$ | 445nm-532nm | 532nm-660nm | 445nm-660nm |
|---|---|---|---|
| $\alpha_{abs}$ | 2.6 ±0.8 | 1.3 ±0.4 | 1.9 ±0.6 |





Table 4:  Single scattering albedo, ω(λ), (nephelometer wavelengths) and scattering Angström exponent, $\alpha_{sca}$, derived for 31 October 2013 (24h average) at Durlacher Tor,  Karlsruhe

| ω(450) | ω (550) | ω(700) | $\alpha_{sca}$ 450-700 |
|--------|---------|--------|------------------------|
| 0.85±0.3 | 0.72±0.2 | 0.78 ±0.3 | 1.7±0.2 |





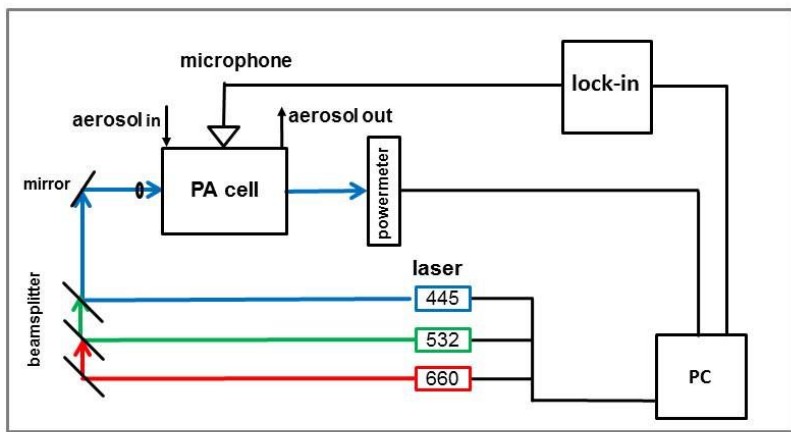

Figure 1: Setup of the 3-wavelength photoacoustic spectrometer





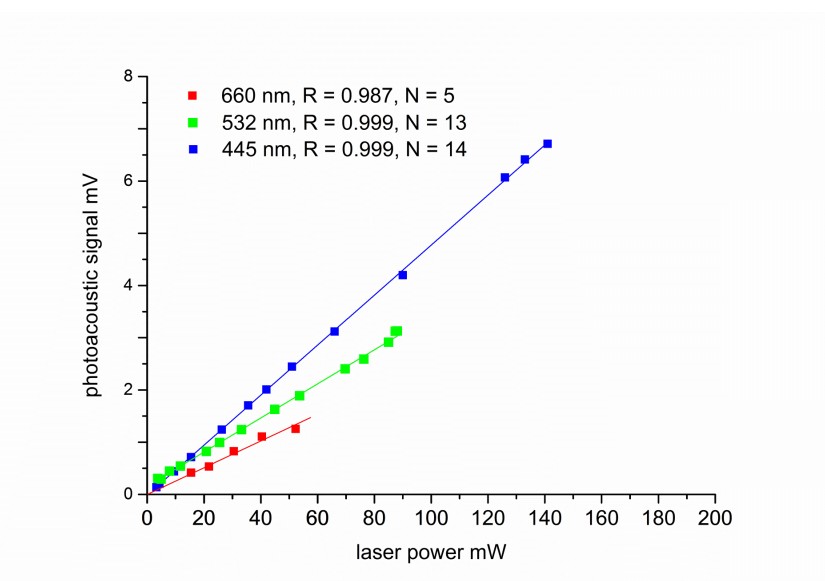

Figure 2: Test of the linearity response of the photoacoustic detection system with laser power





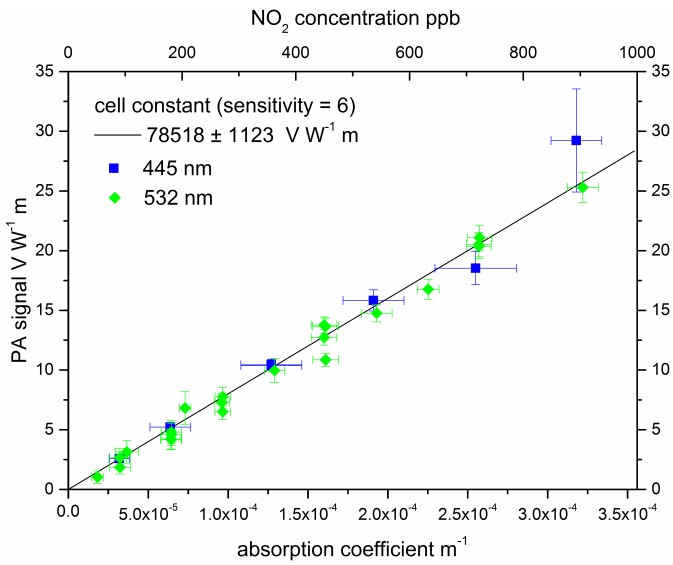

Figure 3: Cell constant determined for the lock-in sensitivity 6 from calibration measurements with $NO_2$ at 445 nm and 532 nm. Note that some of the 532 nm measurements were performed at sensitivity 7 and were subsequently rescaled to sensitivity 6.





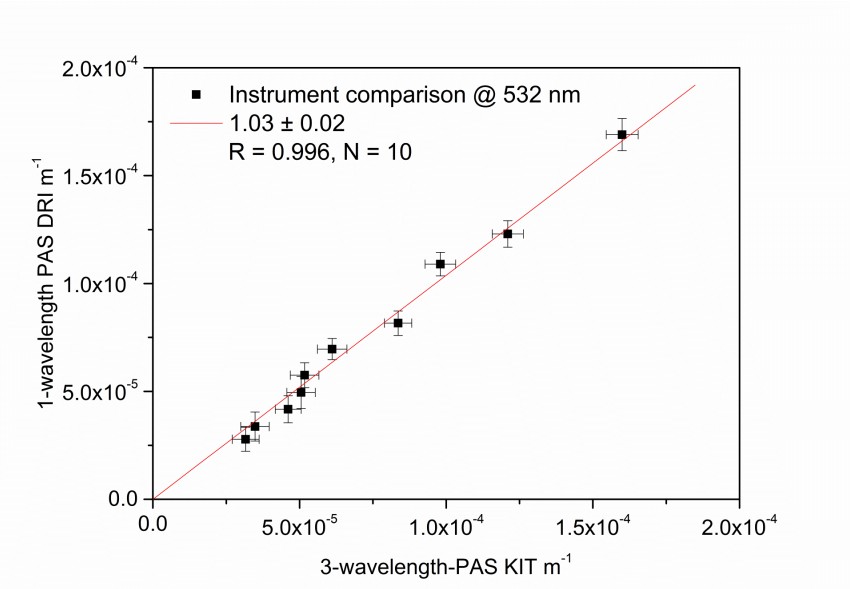

Figure 4: Comparison of two photoacoustic instruments (DRI, PAS KIT) for CAST soot C/O 0.29 at 532nm




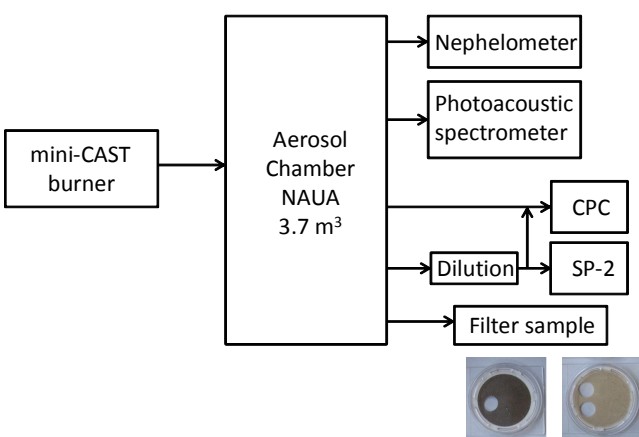

Figure 5: Setup of instruments during the laboratory study SOOT15
and filter samples of CAST soot C/O 0.29 (left) and C/O 0.38 (right)





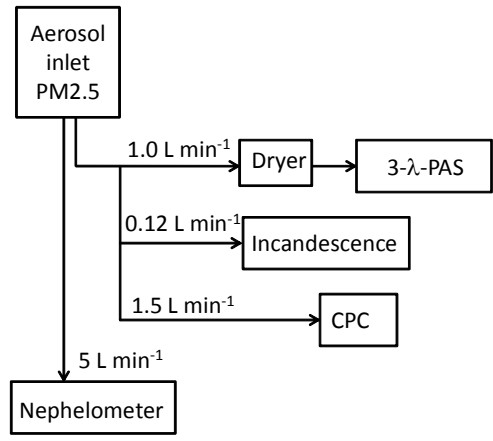

Figure 6: Setup of instruments during field study



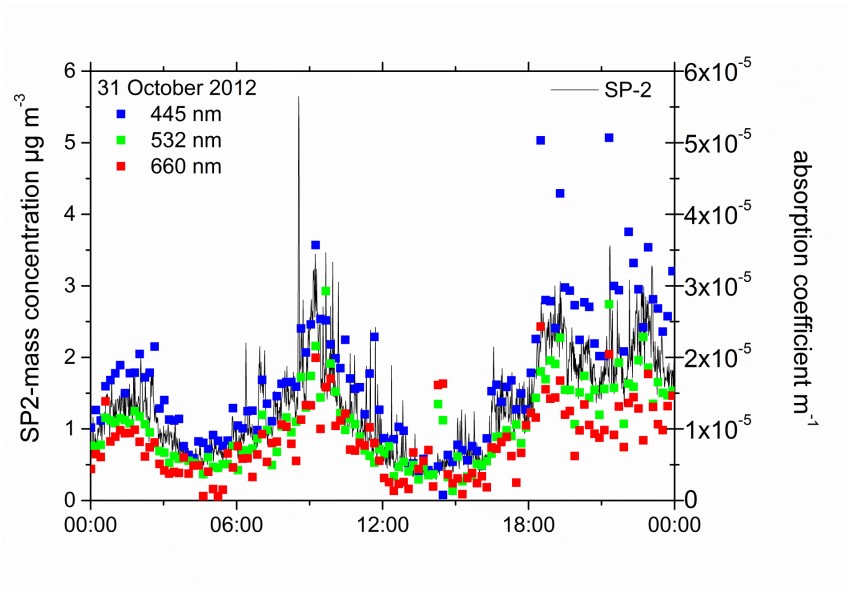

Figure 7: SP-2 incandescence mass concentrations (black line) and photoacoustic
absorption coefficients (blue, green, red squares) for one day in autumn 2012,
Karlsruhe, Germany



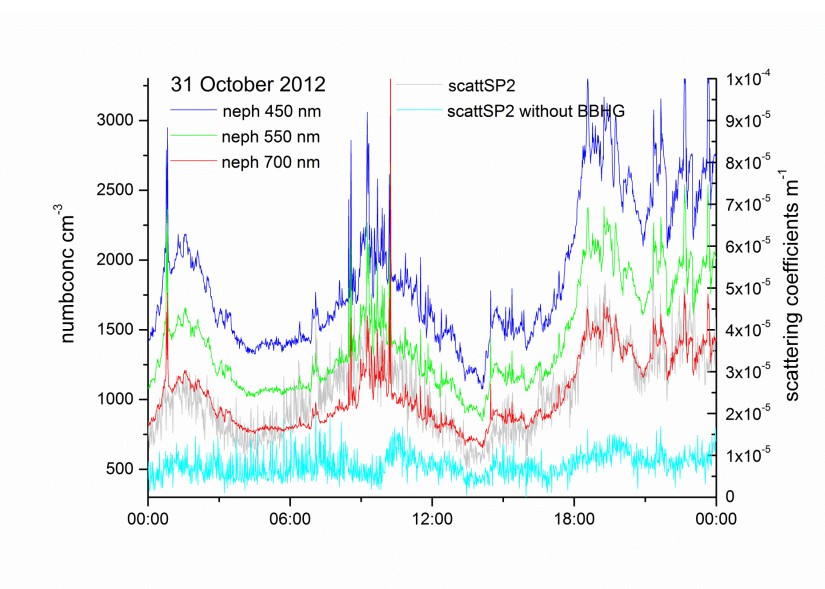

Figure 8: Number concentrations measured by SP-2 for all scattering particles (grey line)
and scattering particles without incandescence (light blue line) compared to
scattering coefficients from Nephelometer at three wavelengths