# Peer review of "A NOVEL SINGLE-CAVITY MULTI-WAVELENGTH PHOTOACOUSTIC SPECTROMETER FOR ATMOSPHERIC AEROSOL RESEARCH"

_Atmospheric Measurement Techniques, 2016_

## Referee Comment (RC1) · Anonymous Referee #1 · 20 Jun 2016

GENERAL

The paper describes the technical details of a multi-wavelength photoacoustic spectrometer and its application to laboratory and field measurements. There are not many photoacoustic spectrometers available, practically only one manufacturer so new instruments are more than welcome to the field. The paper is well written and there are no major mistakes so I can recommend its publication in AMT with a few minor revisions, most of which are just additional details.

In the introduction you should write something of PAS's problems, too, it is not a perfect method either. Especially, it is sensitive to high relative humidity, write something about that with some references (see, e.g., Arnott et al., JGR, 108, 4034,

doi:10.1029/2002JD002165; Raspet et al., J. Atmos. Oceanic Technol., 20, 685–695). And that absorbing gases, mainly NO2 have to be dealt with.

There is one piece of information that would add the value of the results: the size of the rBC core, the mass fraction of rBC, and the thickness of the coating. These affect the MAC and can be obtained from the SP-2 data. If it is not too much work, I would recommend you add this information to your tables and figs and discuss it a bit. Consider it, but it is ok also if you don't add it, after all this is an AMT paper showing methodological development.

DETAILED COMMENTS

P4,L2: "...in the near-UV ..." 445 nm is visible blue light, not UV by any definition.

P5, L12: "... aerosol flow of 1std. liter per minute.." How is this maintained? Mass flow controller or what? Is accurate flow actually important? It does not appear in Eq. (1) at all. How does the flow rate affect particle losses? Did you measure size-dependent particle detection efficiency? If you did, please report the main results.

Section 2. Is there a relative humidity sensor somewhere in the instrument? I did not find an info on such. Considering the sensitivity to RH, it should be measured.

P7,L38 " ... The TC, EC, and OC contents of the aerosol samples were determined from quartz fiber filters.." Describe the sampling method, now there is nothing. At least sampler, size range it is sampling, flow rate, filter type.

P8,L20-21 "... At the beginning and the end of each experiment, filter samples were taken for off-line OC/EC analysis". How stable were the concentrations according to the other instruments' data? This is relevant, since you only sampled at the beginning and end of the experiment. In the results show also time series plots of the chamber experiments and note there the times when the filter samples were taken.

P9, L36-37 "In order to avoid perturbation of the aerosol sampling during the optical measurements, no filter sampling was possible in parallel with the experiments." Why

would filter sampling disturb the experiment?

P10, L35-36 " A reliable SP-2 incandescence measurement at these high C/O ratios was found to be impossible..." Is this due to concentration below SP-2 detection limit or what?

P11,L17-29 " The rBC mass measured by the SP-2 incandescence method was compared to the off-line elemental carbon (EC) and total carbon (TC) analysis results that were obtained by the thermo-optical method." I don't find the results of this comparison. A scatter plot or the EC&OC concentrations in Table 1 would do it.

P11,L31-32 " ... due to the increase in the OC mass, the MAC of TC (MAC-TC) decreases with increasing C/O ratio ..." Where is this shown?

P14,L12 " The trend of the nephelometer data ... " The concept of "trend" is something else. Trend is when something increases or decreases over a longer period of time, here you only show 24 hours of data. Rephrase the sentence.

P14,L14 "...while there is no correlation with the number concentration of rBC-free scattering particles..." This is not quite true. The correlation coefficient sure is lower but when I look at the time series in Fig 8, after about 10:00 the light blue line varies actually fairly nicely the variations of the scattering coefficients. How about adding also the total number concentrations measured with the CPC in the figure? Now you don't use the CPC data anywhere.

In the Tables you have used the symbol sigma for the mass absorption coefficient but MAC in the text. Change either of them to be consistent. And in Table 1, were the mass absorption coefficients calculated from rBC or EC concentrations? Whichever they were, the other could be added there as well, just like in Table 2. Are the results in Table 1 from SOOT11 or SOOT15 or both? Show that somehow in the table.
* * *

---

## Referee Comment (RC2) · Anonymous Referee #2 · 1 Aug 2016

The manuscript by Linke et al. describes a custom-built, three-wavelength photoacoustic instrument. Sufficient technical details are provided, and a generous amount of measurements are presented for instrument validation and demonstration. The manuscript is well written, describes an instrument that would be useful to many atmospheric scientists, and includes some basic measurements of MAC that add to the literature. For these reasons I recommend its publication after the following comments are addressed:

Comment 1:

The manuscript sometimes refers to the three-wavelength photoacoustic as "multi-wavelength". I would say the term "three-wavelength", which is already used in

the manuscript more than once, is much more appropriate, especially in the context of other methods of measuring or estimating absorption, like the 7-wavelength aethalometer.

Comment 2:

This comment is more substantial and relates to the interpretation of the diffusion flame samples. The diffusion flame was operated under various conditions, which resulted in varying % of "OC" being measured, where OC is defined by thermal evolution of carbon in an inert atmosphere.

The manuscript currently discusses this OC % in direct comparison to the OC % of atmospheric aerosols. This is incorrect. The OC that is measured in the particles produced by fuel-rich diffusion flames has been shown to represent incompletely graphitized soot (Maricq 2014). This stands in contrast to the typical OC found in the atmosphere, which forms from "normal" organic material via terpene oxidation, hydrocarbon evaporation, etc. These "normal" organics will be very different from partially graphitized soot, in terms of their volatility, reactivity, light absorption, and hygroscopicity. Therefore, the material currently described as OC is not comparable to atmospheric OC, and the two should not be compared. At the simplest level, a similar OC% between the diffusion flame soot and atmospheric particles clearly does not warrant a comparison of the MAC between such samples.

Of course, the diffusion flame OC is still a reproduceable and well-defined material, so the reported MAC values are likely to be useful to others. But the authors should make this distinction clear, and change the discussion at "Discussion of the chamber results", on page 4 first paragraph, on page 12 second paragraph, and wherever else is relevant.

Note also that the chemical uniqueness of the partially graphitized flame soot also explains why the SP2 did not observe incandescence signals (page 9, line 30). Since the material is incompletely graphitized, what was measured as EC may have formed

by charring during OC/EC analysis, and what was originally present apparently had a MAC that was too low for the SP2 to bring it to incandescence. Alternatively, the material may not have been refractory enough to incandesce.

Comment 3:

On page 7 line 1, the LOD is discussed. What was the averaging interval? What was the influence of longer averaging times? An Allan variance analysis would be the best way to answer these questions. What I am requesting is given by e.g. Onasch et al. (2015).

List of minor comments:

I also have a few minor comments after reading the manuscript carefully.

page 2, line 35 – since BrC typically does not increase indefinitely with decreasing wavelength, please give a number to "shorter wavelengths"

page 3, line 30 here cite Collaud Coen et al. Atmos Meas Tech 2010, 3, 457-474.

page 3, line 41, why is there weak cros sensitivity to particle light scattering?

page 5, line 26, have the authors investigated whether a different transmission efficiency of such gases as NO2 might cause bias in the background measurements? Some gases may interact significantly with the PM filter.

page 6, line 4, is the instrument temperature controlled? Could variations in temperature lead to a bias in the corrected laser power?

page 6, line 29, what does field proven mean? Proven against what reference?

page 10, line 24 and elsewhere. Ideally a quantitative statistical analysis would be given, such as a two-tailed t-test, instead of the statement that values agree nicely or not.

page 12, line 11 Here the reader wonders what the difference between the different

EC/OC protocols is, especially with respect to their charring artifacts (see OC% discussion above). A short comment or citation would be helpful.

page 13, line 36, "home heating season" would be clearer.

page 13, line 37, why were the other periods described if only 31 october is included here?

page 14 line 20, were the limits of sensitivity of the SP2 with respect to particle size corrected for?

Figure 8 is hard to read and includes some labels which are not descriptive, like numbconc, scattSP2 and BBHG.

References

Maricq, M. M.: Examining the Relationship Between Black Carbon and Soot in Flames and Engine Exhaust, Aerosol Sci. Technol., 48, 620–629, doi:10.1080/02786826.2014.904961, 2014.

Onasch, Timothy B., et al. "Single scattering albedo monitor for airborne particulates." Aerosol Science and Technology 49.4 (2015): 267-279.

---

## Referee Comment (RC3) · Anonymous Referee #3 · 1 Aug 2016

This a well written description of a photoacoustic spectrometer and few examples of laboratory and ambient measurements, but there are a few issues that could be discussed in more detail.

1) A detailed discussion of accuracy. I expected to see discussion of possible systematic biases and overall determination of accuracy at high signal to noise.

2) How is the resonant frequency determined? Calculated based on the cell geometry? is it set such that there is 90 deg. Phase between the light and mic signal? What is the acoustic amplification, Q, of the cell?

3) What is the level of the bkg in units of absorption? How stable is the background? Is

the background due to absorption in the windows? Or scattered light absorbed in the cell walls?

4) Absorption of PAS light by NO2 will convert some for the NO2 to NO. This effect will bias the calibration. Could the authors quantify the magnitude of this effect? It is important?

5) Can you describe the lockin amplifier gain in a more meaningful manner. Settings 6 and 7 do not mean much to the reader.

---

## Author Comment (AC1) · 9 Aug 2016

**We thank Referee#1 for the positive comment. These comments helped to substantially improve the manuscript. Below we give detailed answers to the individual reviewer comments in blue.**

RC#1: In the introduction you should write something of PAS's problems, too, it is not a perfect method either.

Answer: Regarding the limitations of the PA technique we added to the introduction on P3,L43:

*"However, despite these advantages of a direct absorption measurement, during field applications the presence of ambient trace gases, like NO2, ozone or water vapor can influence the photoacoustic measurement (Arnott et al. 1999). Especially the impact of high relative humidities on the photoacoustic measurement has to be considered (Arnott et al. 2003, Raspet et al. 2003)."*

There is one piece of information that would add the value of the results: the sizeof the rBC core, the mass fraction of rBC, and the thickness of the coating. These affect the MAC and can be obtained from the SP-2 data. If it is not too much work, I would recommend you add this information to your tables and figs and discuss it a bit. Consider it, but it is ok also if you don't add it, after all this is an AMT paper showing methodological development.

We considered it, but we decide against it.

DETAILED COMMENTS

P4,L2: "...in the near-UV ..." 445 nm is visible blue light, not UV by any definition.
*Changed into: " in the visible spectral range"*

P5, L12: "... aerosol flow of 1std. liter per minute.." How is this maintained? Mass flow controller or what? Is accurate flow actually important? It does not appear in Eq. (1) at all. How does the flow rate affect particle losses? Did you measure size-dependent particle detection efficiency? If you did, please report the main results.
Section 2. Is there a relative humidity sensor somewhere in the instrument? I did not find an info on such. Considering the sensitivity to RH, it should be measured.

Answer:The aerosol flow is maintained by a mass flow controller. An accurate flow rate is not important for the photoacoustic measurement, but it was given with respect to the exchange time of the cell volume. We did not explicitly measure the size-dependent particle detection efficiency, but estimated the size-segregated aerosol transmission.

We inserted in P5, L12:
*"A continuous aerosol flow of 1 std. liter per minute through the cell is maintained by a mass flow controller (Mykrolis, Tylon 2900 series). At this flow rate the calculated cell volume of about 265 cm3 is exchanged around 3 to 4 times a minute. The calculated aerosol transmission efficiency through the cell including the acoustic notch filters is 97% for particles with a size of 1 µm and a density of 1.8 g/cm3."*

During chamber experiments the chamber was filled with dry synthetic air. During field measurements we used a drier upstream the PAS and nephelometer. The RH values recorded by the nephelometer varied between 40-60% RH during the campaign period.

We inserted: *"During the laboratory chamber studies the RH was always below 30 %. During field measurements there was a silica gel drier installed upstream the PA and nephelometer sampling line, which confined the RH to below 60 % throughout the campaign."*

P7,L38 " ... The TC, EC, and OC contents of the aerosol samples were determined from quartz fiber filters.." Describe the sampling method, now there is nothing. At least sampler, size range it is sampling, flow rate, filter type.

Changed to: "
*"The TC, EC, and OC concentrations of the aerosol samples were determined from thermal analyses of particle laden quartz fiber filters. For the filter sampling, 47 mm diameter quartz fiber filters (Munktell MK360) were inserted into a stainless steel filter holder (Satorius) and connected directly to the aerosol chamber. With the aim of a mass flow controller with a set flow of 10 std. L min-1 a defined volume of aerosol was sampled from the chamber through the filter. TC, EC and OC analysis of the particle laden filters were performed using a Sunset OC/EC thermal analyzer (Sunset Laboratory Inc., USA) by applying the EUSAAR-2 temperature protocol (Cavalli et al. 2010)."*

P8,L20-21 "... At the beginning and the end of each experiment, filter samples were taken for off-line OC/EC analysis". How stable were the concentrations according to the other instruments' data? This is relevant, since you only sampled at the beginning and end of the experiment. In the results show also time series plots of the chamber experiments and note there the times when the filter samples were taken.

We added another plot of the chamber measurements during the SOOT11 campaign, which shows SP2 and EC mass concentrations as well as PAS mass concentrations calculated from EC-specific MAC.

P9, L36-37 "In order to avoid perturbation of the aerosol sampling during the optical measurements, no filter sampling was possible in parallel with the experiments." Why would filter sampling disturb the experiment?

The SOOT11 campaign took place at the AIDA chamber with pressure control, while the SOOT15 campaign was performed in the much smaller NAUA chamber. At the time of the SOOT15 campaign, the NAUA chamber was not equipped with an automated pressure control to balance additional differential pressures as they are induced during filter sampling.

P10, L35-36 " A reliable SP-2 incandescence measurement at these high C/O ratios was found to be impossible..." Is this due to concentration below SP-2 detection limit or what?

"*Yes, concentrations below the SP-2 detection limit*"

P11,L17-29 " The rBC mass measured by the SP-2 incandescence method was compared to the off-line elemental carbon (EC) and total carbon (TC) analysis results that were obtained by the thermo-optical method." I don't find the results of this comparison. A scatter plot or the EC&OC concentrations in Table 1 would do it.

P10, L17-29:
"The explanation in this part was confusing, so we inserted another table which gives an overview of the different comparisons we have done".

P11,L31-32 " ... due to the increase in the OC mass, the MAC of TC (MAC-TC) decreases with increasing C/O ratio ..." Where is this shown?

This relates to the results of Schnaiter et al. (2006)

P14,L12 " The trend of the nephelometer data ... " The concept of "trend" is something else. Trend is when something increases or decreases over a longer period of time, here you only show 24 hours of data. Rephrase the sentence.

Changed into:„The temporal evolution of the nephelometer data…".

P14,L14 "...while there is no correlation with the number concentration of rBC-free scattering particles..." This is not quite true. The correlation coefficient sure is lower but when I look at the time series in Fig 8, after about 10:00 the light blue line varies actually fairly nicely the variations of the scattering coefficients. How about adding also the total number concentrations measured with the CPC in the figure? Now you don't use the CPC data anywhere.

"We changed figure 8 and added the CPC concentration"

---

## Author Comment (AC2) · 20 Sep 2016

**We thank Referee#2 for the helpful comments. These comments helped to substantially improve the manuscript. Below we give detailed answers to the individual reviewer comments in blue.**

Comment 1: The manuscript sometimes refers to the three-wavelength photoacoustic as "multiwavelength". I would say the term "three-wavelength", which is already used in the manuscript more than once, is much more appropriate, especially in the context of other methods of measuring or estimating absorption, like the 7-wavelength aethalometer.

**We replaced multi- by three-wavelength photoacoustic spectrometer throughout the manuscript.**

Comment 2: This comment is more substantial and relates to the interpretation of the diffusion flame samples. The diffusion flame was operated under various conditions, which resulted in varying % of "OC" being measured, where OC is defined by thermal evolution of carbon in an inert atmosphere.

The manuscript currently discusses this OC % in direct comparison to the OC % of atmospheric aerosols. This is incorrect. The OC that is measured in the particles produced by fuel-rich diffusion flames has been shown to represent incompletely graphitized soot (Maricq 2014). This stands in contrast to the typical OC found in the atmosphere, which forms from "normal" organic material via terpene oxidation, hydrocarbon evaporation, etc. These "normal" organics will be very different from partially graphitized soot, in terms of their volatility, reactivity, light absorption, and hygroscopicity. Therefore, the material currently described as OC is not comparable to atmospheric OC, and the two should not be compared. At the simplest level, a similar OC% between the diffusion flame soot and atmospheric particles clearly does not warrant a comparison of the MAC between such samples.

Of course, the diffusion flame OC is still a reproduceable and well-defined material, so the reported MAC values are likely to be useful to others. But the authors should make this distinction clear, and change the discussion at "Discussion of the chamber results", on page 4 first paragraph, on page 12 second paragraph, and wherever else is relevant.

Note also that the chemical uniqueness of the partially graphitized flame soot also explains why the SP2 did not observe incandescence signals (page 9, line 30). Since the material is incompletely graphitized, what was measured as EC may have formed by charring during OC/EC analysis, and what was originally present apparently had a MAC that was too low for the SP2 to bring it to incandescence. Alternatively, the material may not have been refractory enough to incandesce.

This is indeed an important point and we revised the manuscript as described in the following to make the difference between flame soot OC and atmospheric OC clearer.

P4, first paragraph: At this point we have a general introduction of the paper's structure. In connection with the soot generation we used the term "surrogates", without a validation of the comparability to atmospheric carbonaceous particles. This might be somewhat misleading to the reader and, therefore, we changed "... carbonaceous particles with different OC content" to "... combustion particles with different OC content" to be more precise.

In the discussion of the chamber experiments, P11 of the original manuscript, we added: "It is important to note that the CAST propane diffusion flame generator was used as a reproducible source for analogue combustion aerosol. By changing the fuel-to-oxygen ratio, the influence of incomplete combustion – manifested by increasing organic carbon (OC) content – on the absorbing properties of the soot aerosol could be systematically investigated. However, combustion OC material does not necessarily represent all atmospheric OC compounds as most of this material stems from secondary processes like the oxidation of terpenes. This should be kept in mind, when comparing our laboratory results with results from field measurements."

On P12, L42 we replaced "This finding relates to results obtained in field measurements by Kondo ..." with:

"Even though these laboratory studies are not directly comparable to atmospherically processed combustion emissions and, furthermore, different methods were used in the laboratory and field studies, it should be mentioned that comparable MAC values of BC are also deduced from atmospheric measurements. Kondo et al...."

Comment 3: On page 7 line 1, the LOD is discussed. What was the averaging interval? What was the influence of longer averaging times? An Allan variance analysis would be the best way to answer these questions. What I am requesting is given by e.g. Onasch et al. (2015).

The averaging interval was 20 s. We will act on the suggestion to investigate the influence of longer averaging times.

**List of minor comments:**

I also have a few minor comments after reading the manuscript carefully.

Page 2, line 35 – since BrC typically does not increase indefinitely with decreasing wavelength, please give a number to "shorter wavelengths"

**P2, L35 was changed to:**

"For BC, the imaginary part of the refractive index is nearly wavelength independent over the visible and near-UV spectral range. In contrast, the imaginary part of the refractive index of brown carbon (brC) continuously increases from the red over the blue to at least the near-UV spectral range."

Page 3, line 30 here cite Collaud Coen et al. Atmos Meas Tech 2010, 3, 457-474.

On P3, L31 we added: (Collaud Coen et al. 2010, Lack et al. 2008)

Page 3, line 41, why is there weak cross sensitivity to particle light scattering?

On P3, L41 we removed: "... or only weak ..."

Page 5, line 26, have the authors investigated whether a different transmission efficiency of such gases as NO2 might cause bias in the background measurements? Some gases may interact significantly with the PM filter.

Until now, we haven't found any indications for a substantial interaction of gaseous species with the background filter material that causes a detectable bias.

Page 6, line 4, is the instrument temperature controlled? Could variations in temperature lead to a bias in the corrected laser power?

No, the instrument is not temperature controlled. We agree with the Reviewer that variations

in the temperature can result in a drift of the characteristics of the acoustic resonator (but not in the laser power measurement as this done with a thermopile detector that is less prone to drifts in ambient temperature). The impact of a temperature drift onto the cell constant of the resonator and, consequently, the measured absorption coefficient is currently being investigated. However, the acoustic resonator has a low enough quality function, so that a drift of a few degrees should not result in a significant change of the cell constant.

Page 6, line 29, what does field proven mean? Proven against what reference?

On P6, L29 we removed: field-proven

Page 10, line 24 and elsewhere. Ideally a quantitative statistical analysis would be given, such as a two-tailed t-test, instead of the statement that values agree nicely or not.

We compared the results from two different campaigns where different methods and instruments were used. So, a qualitative statement on the comparability of the results is justified as otherwise a comprehensive discussion of all instrumental and methodical errors is necessary which is certainly beyond the scope of this manuscript.

Page 12, line 11 Here the reader wonders what the difference between the different EC/OC protocols is, especially with respect to their charring artifacts (see OC% discussion above). A short comment or citation would be helpful.

We added citation of the following paper to the manuscript: Watson et al., Aerosol and Air Quality Research, Vol. 5, No.1, pp. 65-102, 2005

Page 13, line 36, "home heating season" would be clearer.

P13, L36 was changed to: home heating season

Page 13, line 37, why were the other periods described if only 31 October is included here?

Here we want to give a short description of the overall weather situation, and especially the change in the conditions from warm to cold, which we believe is important for the observed change in the wavelength-dependent aerosol absorption.

Page 14 line 20, were the limits of sensitivity of the SP2 with respect to particle size corrected for?

No, these sensitivity limitations are not corrected.

Figure 8 is hard to read and includes some labels which are not descriptive, like numbconc, scattSP2 and BBHG.

Figure 8 is changed in the revised manuscript.

---

## Author Comment (AC3) · 20 Sep 2016

**We thank Referee#3 for the helpful comments. These comments helped to substantially improve the manuscript. Below we give detailed answers to the individual reviewer comments in blue.**

This is a well written description of a photoacoustic spectrometer and few examples of laboratory and ambient measurements, but there are a few issues that could be discussed in more detail.

1) A detailed discussion of accuracy. I expected to see discussion of possible systematic biases and overall determination of accuracy at high signal to noise.

The determination of the characteristics of a prototype instrument in terms of accuracy, systematic biases, and detection limit is an ongoing process and needs the comparison of the instrument with other more established measurement methods and instruments. This is still ongoing work, along with further instrument modifications and improvements.

All information on the instrument characteristics that we could gather so far are already compiled in the paper. There was only one occasion to validate our instrument against the established single wavelength photoacoustic instrument (DRI-PAS), originally developed by Pat Arnott from the Desert Research Institute. An intercomparison study to test the accuracy of our prototype instrument (at 532 nm) was performed with CAST soot aerosol. The result is shown in Figure 4 of the manuscript.

We have planned further studies in this respect, e.g. by comparing our instrument with the extinction minus scattering method established at the World Calibration Center for Aerosol Physics (WCCAP) in Leipzig, Germany. Furthermore, we will validate our $NO_2$ calibration method by a simultaneous extinction measurement across the acoustic resonator length (in this way being insensitive to any variations in laser and $NO_2$ spectral properties). The results of all these activities will be the subject of forthcoming publications.

2) How is the resonant frequency determined? Calculated based on the cell geometry? Is it set such that there is 90 deg. Phase between the light and mic signal? What is the acoustic amplification, Q, of the cell?

The resonance frequency was determined each day by a scan in the applied frequency range. The Q-factor of the cell is about 25.

3) What is the level of the bkg in units of absorption? How stable is the background? Is the background due to absorption in the windows? Or scattered light absorbed in the cell walls?

The LOD is given in units of absorption coefficient.
The background is generated due to interaction of windows and cell wall with scattered laser light resulting in a background noise. This noise also depends from the surrounding area.

4) Absorption of PAS light by NO2 will convert some for the NO2 to NO. This effect will bias the calibration. Could the authors quantify the magnitude of this effect? It is important?

For the shortest wavelength we used in this study, i.e. 445 nm, we didn't find any indications for photo dissociation of NO2. This is in agreement with literature data of NO2 quantum yields (e.g. by Roehl et al., 1994). However, we agree that this is an important aspect for our calibration procedure in conjunction with future setups using even shorter wavelengths. (Roehl et al., J. Phys. Chem., 98, 7837-7843, 1994)

5) Can you describe the lock in amplifier gain in a more meaningful manner? Settings 6 and 7 do not mean much to the reader.

These numbers correspond to sensitivity setting for full scale (= 10 V) output. The settings in the ultra-stable mode of 6 and 7 correspond to sensitivities of 1mV/10nA and 300µV/3nA, respectively.